# Methane Emission, Carbon Footprint and Productivity of Specialized Dairy Cows Supplemented with Bitter Cassava (*Manihot esculenta* Crantz)

**DOI:** 10.3390/ani14010019

**Published:** 2023-12-20

**Authors:** Isabel Cristina Molina-Botero, Xiomara Gaviria-Uribe, Juan Pablo Rios-Betancur, Manuela Medina-Campuzano, Mercedes Toro-Trujillo, Ricardo González-Quintero, Bernardo Ospina, Jacobo Arango

**Affiliations:** 1Tropical Forages Program, International Center for Tropical Agriculture (CIAT), Km 17, Palmira 763022, Valle del Cauca, Colombia; isamo2609@gmail.com (I.C.M.-B.); xgavuribe@gmail.com (X.G.-U.); r.gonzalez@cgiar.org (R.G.-Q.); 2Colanta, Calle 74# 64ª-51, Medellín 050044, Antioquia, Colombia; jpriosb@unal.edu.co (J.P.R.-B.) ; manuelamc@colanta.com.co (M.M.-C.); mercedestt@colanta.com.co (M.T.-T.); 3Corporacion Clayuca, International Center for Tropical Agriculture (CIAT), Km 17, Palmira 763022, Valle del Cauca, Colombia; b.ospina@clayuca.org

**Keywords:** greenhouse gases, milk production, ruminants, supplementation

## Abstract

**Simple Summary:**

The objective of this research was to determine the effect of cassava (*Manihot esculenta* Crantz) supplementation on enteric methane emissions, carbon footprint, and production parameters in dairy cows. Cassava roots and leaves replaced up to 30% of the daily supply of commercial concentrate for two Jersey and Jersey * Holstein breeds. Cassava leaves were characterized by a high crude protein content, with five times more neutral detergent fiber content than cassava root. Average enteric methane emissions per animal ranged from 194 to 234 g/d. The carbon footprint was reduced by replacing 30% of the concentrate with cassava leaves and/or roots. Energy-corrected milk production was 1.15 times higher in Jersey * Holstein animals than in Jersey cows. Therefore, supplementation with cassava leaves and/or roots is a nutritionally and environmentally sustainable strategy to replace external grain concentrates used in these systems.

**Abstract:**

The objective of this research was to determine the effect of cassava (*Manihot esculenta* Crantz) supplementation on enteric methane (CH_4_) emissions, carbon footprint, and production parameters in dairy cows. Daily concentrate supply for Jersey and Jersey * Holstein breeds was evaluated in four treatments (T): T1: 100% commercial concentrate; T2: 70% concentrate + 30% cassava leaves; T3: 70% concentrate + 30% cassava roots; and T4: 70% concentrate + 15% cassava leaves + 15% cassava root chips. Measurements of CH_4_ emissions were performed using the polytunnel technique. Average daily dry matter intake ranged from 7.8 to 8.5 kg dry matter (DM). Cassava leaves were characterized by a high crude protein (CP) content (171 g CP/kg DM), with 5 times more neutral detergent fiber (NDF) content than cassava root (587 vs. 108 g NDF/kg DM). Average enteric CH_4_ emissions per animal ranged from 194 to 234 g/d (*p* > *0.05*). The carbon footprint was reduced by replacing 30% of the concentrate with cassava leaves and/or roots. Energy-corrected milk production was 1.15 times higher in Jersey * Holstein animals than Jersey cows (47 vs. 55 kg). Therefore, supplementation with cassava leaves and/or roots is a nutritionally and environmentally sustainable strategy.

## 1. Introduction

Globally, methane (CH_4_) is the second most important greenhouse gas (GHG) emitted by the agricultural sector, mainly because it has a global warming potential 28 times higher than that of carbon dioxide [1]. However, its permanence in the atmosphere is lower [2]. Additionally, developing countries emit 70% of the total global enteric CH_4_, of which 25% comes from Latin America and the Caribbean [3]. In livestock systems, CH_4_ is generated as a product of enteric fermentation and is emitted to the environment mainly through belching, which represents a significant energy loss for the animal [4]. 

The carbon footprint is the sum of CH_4_, nitrous oxide (N_2_O), carbon dioxide (CO_2_) and other gases emitted directly or indirectly during the process of obtaining a product [5]. However, it has been stated that the carbon footprint per gram of protein in milk or meat from ruminants is very high compared to other meats, such as pork, poultry, or rabbit, or other animal products such as yogurt or eggs [6]. For the case of ruminants, the amount of gases in the carbon footprint is directly related to the type of feed consumed, [7] and indirectly a reduction in the carbon footprint of milk production has been established when animal performance and farm profitability are improved [8,9].

In recent years, the livestock sector, as well as other economic sectors, has made great efforts in the search for and implementation of strategies to reduce GHG emissions, particularly of CH_4_. The sector aims to achieve sustainable production systems that contribute to the national commitments made before the United National Framework Convention on Climate Change (UNFCCC). Thus, several options have been evaluated to mitigate enteric CH_4_ emissions [10], among which are feeding management, feed composition, forage quality, modifications of the microbial community, chemical manipulation, and animal crossbreeding [11,12,13].

Despite knowledge of these strategies, the Colombian dairy sector resists change due to historically low profit margins. Milk prices hardly compensate for the cost of raw materials such as fertilizers, corn, and soybean used in the production of concentrates [14]. Adoption of GHG mitigation strategies will be low if they cannot also increase productivity in dairy systems, reduce dependence on external inputs, increase producers’ profits, and make sustainable use of natural resources. In this sense, GHG mitigation actions must allow for the partial or total replacement of grains in concentrate feed formulations with nutrition sources that cost less and are locally available [15].

In the search for nutritional alternatives, research has been conducted on cassava (*Manihot esculenta* Crantz) as a supplement for ruminants. This tropical and subtropical shrubby plant, belonging to the Euphorbiaceae family, is characterized by a high tolerance for poor soils and adverse climatic conditions [16]. Supplementation with cassava in ruminant feed has been used for its nutritional value [17,18]. Among other qualities, roots have a high content of non-structural carbohydrates (75 to 85%) and low levels of crude protein (CP; 2–3% CP), while leaves and green stems contain higher CP values (25%) [16,19]. Additionally, multiple authors found antimethanogenic compounds such as tannins and saponins within cassava roots, stems, and leaves [1,20,21].

Despite cassava’s importance—Colombia produces approximately 2.1 million tons per year on 187.2 thousand hectares [22]—no field-based studies have been carried out on the effect of cassava intake on the combination of productive parameters, carbon footprint, and enteric CH_4_ emissions in cattle or the relationship of these parameters with the most common cattle breeds and crosses of specialized dairy cows. 

Furthermore, there appears to be a correlation between animal production parameters and genetic characteristics, although the nature of this relationship is complex. Extensive research is essential to comprehensively understand these correlations over time, considering various production systems and environmental conditions [23]. Notably, crossbreeding has been suggested as a potential avenue for enhancing performance and mitigating CH_4_ emissions. Therefore, it is crucial to assess the CH_4_ emissions of different breeds and crosses commonly employed in specialized dairy farming to ascertain the genetic connections between methane production and existing selection traits [24].

Considering the aforementioned factors, this study represents the inaugural investigation conducted under field conditions with specialized dairy cows supplemented with cassava in Colombia. The primary aim was to assess the impact of supplementation with cassava leaves and roots on CH_4_ emissions, the carbon footprint, and various key productivity parameters.

## 2. Materials and Methods

### 2.1. Experiment Location

The study was conducted at the Los Cerezales farm, located in the municipality of San Pedro de los Milagros (6°27′0″ N 75°33′0″ W) in the northern region of the department of Antioquia (Colombia), at an altitude of 2350 m above sea level. The area is characterized by soils of the Andisol order, an average annual temperature of 14.2 °C, average annual precipitation of 1714 mm, and a relative humidity of 79% [25].

### 2.2. Treatments Evaluated/Assessed

The treatments evaluated were composed of kikuyu grass (*Cenchrus clandestinus (Hochst. ex Chiov.) Morrone*) as a base forage and a commercial concentrate for dairy cows in production. The supply of cassava replaced the daily amount of concentrate in the following proportions: Treatment 1 (T1)—control: 100% commercial concentrate; Treatment 2 (T2): 70% concentrate + 30% cassava leaves; Treatment 3 (T3): 70% concentrate + 30% cassava root meal in the form of chips; and Treatment 4 (T4): 70% concentrate + 15% cassava leaves + 15% cassava chips. 

A preliminary test was conducted two months prior to the experimental period, involving 10 randomly selected animals from the study farm. The objective was to determine the optimal percentage of concentrate replacement with cassava supplements. The animals’ consumption was assessed by substituting the commercial concentrate with varying percentages of cassava leaves and chips—specifically, 20, 30, 40, and 50%. The outcomes of this test indicated that the maximum substitution percentage should be set at 30%, as the animals consumed the entire offered amount within a short duration during the two daily milking procedures. The cassava leaves and chips were harvested in the municipality of San Antonio de Palmito (Department of Sucre), with a regrowth age of 4.5 months. They were air-dried for subsequent transfer and supply to the animals in the tested farm.

### 2.3. Animal Characteristics

In the current study, two distinct breed groups were assessed as experimental groups. The initial group comprised four Jersey cows, averaging 3.3 ± 0.5 years in age, 410 kg in live weight, 2.5 ± 0.9 calvings, 31.8 ± 15.3 days into lactation, and an initial milk production of 18 ± 2.8 L/d. The second group comprised four F1 cows (Jersey * Holstein), averaging 3.7 ± 0.8 years in age, 500 kg in live weight, 2.3 ± 0.4 calvings, 42.8 ± 12.1 days into lactation, with each animal producing an average of 24 ± 4 L/d.

### 2.4. Experimental Design and Animal Management 

The experiment employed a double change-over design, conducted across four consecutive experimental periods for each cow group. During each experimental period, all four treatments were simultaneously evaluated. Throughout the study, the animals enjoyed ad libitum access to water, salt, and kikuyu grass. Daily milking occurred mechanically in both the morning and afternoon, during which each cow received the assigned supplements based on the treatment designated for her group.

The initial 15 days of each period were exclusively allocated for treatment adaptation. Throughout this period, the cows grazed freely in pasture with unrestricted access to kikuyu grass. Subsequently, following this adaptation phase, the cows were relocated to a polytunnel for a 3-day acclimation period. Within the polytunnel, they were provided with fresh kikuyu grass ad libitum. During these 3 days, the cows underwent brief confinement periods within the polytunnel during the day, exiting only at milking time, following the same management protocol outlined earlier. The CH_4_ emission measurements were ultimately recorded on day 19 of each experimental period, spanning a total duration of 152 experimental days for each cow group (Figure 1).

### 2.5. Nutritional Quality of the Treatments

The different components of the treatments (grass, concentrate, leaves, and cassava root) and the collected feces were analyzed to determine their chemical composition and nutritional value at the Chemical and Bromatological Analysis Laboratory of the National University of Colombia and at the Forage Quality and Animal Nutrition Laboratory of the International Center for Tropical Agriculture (CIAT). The collected samples were dried at 55 °C for 72 h, following International Organization for Standardization (ISO) method 6496 to determine dry matter (DM) content [26]. Ash content was quantified by direct calcination in a muffle furnace according to method 942.05 [27]. CP content was determined by the Kjeldahl methodology (CP = N concentration * 6.25). Method 984.13. AOAC, [28]. Neutral detergent fiber (NDF) and acid detergent fiber (ADF) contents were determined using the methodologies proposed by Van Soest et al. [29], adapted to an Ankom Fiber Analyzer AN 3805 (Ankom^®^ Technology Corp., Macedon, NY, USA). Gross calorific value was determined by calorimetry according to ISO 9831 specifications [30]. The starch content in the cassava root and in the commercial concentrate was quantified by enzymatic hydrolysis (Batey in 1986) modified by Mestres [31] in the Postharvest Quality Laboratory of the CIAT Cassava program. Nutrient digestibility of the diets was determined as the ratio of the material consumed to the material excreted. Total feces were estimated from the equation published by Nennich et al. [32].
$$\mathrm{Total}\;\mathrm{feces}=\left[\mathrm{DMI}\;\times \;0.356\;\left(\pm 0.011\right)\right]+0.80\;\left(\pm 0.34\right)$$
where DMI is the daily dry matter intake (kg/d).

### 2.6. Polytunnel Conditions and Quantification of Methane Emissions

Measurements of CH_4_ emissions were performed using the polytunnel technique, as described by Lockyer [33] and Murray et al. [34]. The experiment had two polytunnels, each with a total area of 24 m^2^ and a volume of 62.74 m^3^. Likewise, each polytunnel was divided into two hermetic chambers where one animal per chamber was housed. Each polytunnel chamber had a front entrance for the entrance of the animals and was equipped with a feeder, drinker, and salting trough, plus a fan to facilitate the mixing of gases inside the polytunnel, a thermohygrometer, and an air extractor located on the side opposite the entrance of the animals to remove the air from the polytunnels at a speed of 0.9 m^3^/s.

The polytunnels were completely closed for one hour to accumulate the gas. After this time, the extractor was turned on, and a gas sample was taken directly from the expelled air in each of the polytunnel chambers. The air was collected in a syringe and stored in 10 mL Exetainer^®^ vials until further analysis was performed by gas chromatography. After each measurement, the polytunnel was opened for 5 min. This process was repeated every hour for 24 continuous hours for animals in each measurement period. In addition, every hour an air sample was taken from the outside environment, and the ambient temperature and humidity were recorded. 

The collected gas samples were sent to the Greenhouse Gas Laboratory at CIAT, where their CH_4_ concentration was determined using a Shimadzu GC-2014 gas chromatograph (Shimadzu^®^, Kyoto, Japan), equipped with a flame ionization detector. The ideal gas law was used to calculate the amount of CH_4_ emitted [35], based on the concentration reported by the chromatograph (millimoles) and the total volume of the polytunnel. The amount of CH_4_ emitted by each animal was corrected by the amount of CH_4_ estimated in the environment each hour.

### 2.7. Dry Matter and Nutrient Consumption

Kikuyu grass was offered individually to the cows in feeders installed inside the polytunnels. All animals had free access to grass, salt, and water throughout the experimental period. The kikuyu grass was cut directly from the paddocks where the cows were grazing and was offered fresh and unchopped. The cutting times coincided with the times when the cows would normally consume the grass. The cutting height was set to simulate the consumption behavior observed when the animals were grazing. The voluntary grass consumption of each cow was calculated as the difference between the amounts of grass offered and rejected the following day. Concentrate and supplement were offered to the cows at milking times, individually and in the amounts established for each cow. In no case, throughout the entire experimental period, were rejections of concentrate or supplement observed. Therefore, the amount offered is taken as the total consumption of concentrate and supplement. 

### 2.8. Milk Production and Quality 

Milk production of each animal was recorded weekly throughout the experimental period. On the day of the CH_4_ measurements of each period, milk samples were taken from each cow. Their components were analyzed in the Milk Quality Laboratory with the support of Colanta’s^®^ Dairy Control program. Each sample evaluated was stored in 5 mL containers and transported to the laboratory at a temperature of 4 °C. Fat and protein contents were determined using a Fourier infrared analyzer (MilkoScan FT6000; Foss Analytical^®^ A/S, Hillerod, Denmark), while milk urea nitrogen (MUN) concentration was determined according to method 14637 [36]. Total milk yield was corrected for fat and protein content (3.5 and 3.2%) according to Tyrrell and Reid [37].

### 2.9. Carbon Footprint Estimation/Estimate

Carbon footprint calculations were performed according to the methodology specified by González-Quintero et al. [5] for a specialized dairy system. Briefly, the estimation of greenhouse gas (GHG) emissions and their intensity (emissions expressed per unit of product) were conducted under the life cycle assessment (LCA) methodology. In this study, we successfully employed the measured Tier 3 emission factors for dairy cows as defined by the IPCC [38,39]

The LCA system boundary was defined by the environmental impacts related to specialized dairy farms using a “cradle-to-farm-gate” perspective. Table 1 presents the input data utilized by the two dairy herds. Input quantities were considered for estimating primary GHG emissions, commonly referred to as direct emissions, arising from on-farm usage. Additionally, the table includes secondary GHG emissions (off-farm) originating from the manufacturing and transportation of inputs. It is important to note that no electric energy consumption associated with the milk production process was reported.

To convert CH_4_ and N_2_O emissions to carbon dioxide equivalents (CO_2_eq), the warming power for a time horizon of 100 years was used: 28 for CH_4_ and for 265 N_2_O [1]. The main product of the farms was milk. However, because the farms generate live weight through weaned calves and cull cows, a biophysical allocation rule defined by the International Dairy Federation was used to distribute the absolute emissions between milk and live weight produced [40]. The functional unit used corresponded to 1 kg of fat–protein corrected milk (FPCM) and 1 kg liveweight gain. Calculations were based on real farm data such as herd structure, feeding, pasture management practices and land use, and productive and reproductive information, among others.

### 2.10. Data Analysis

The effect of supplements on CH_4_ production and production variables was determined using the PROC MIXED procedure of the SAS^®^ software, version 9.4 [41]. The separation of means was performed by Tukey’s test with a significance level (α) of 0.05 using the following model:Y*_ijkr_* = *μ* + δ*_i_* +R*_r_*+ P*_j_* + β*_k_* + (δ × R)*_ir_* + ҽ*_ijkr_*
where Y*_ijkr_* is the observation of subject *k* under treatment *i* in period *j* of race *r*; *μ* is the overall mean of the population; δ*_i_* is the effect of the *i*-th treatment (*i* = 1 ..., 4); R*_r_* is the effect of the *r*-th race (*r* = 1, 2); P*_j_* is the effect of the *j*-th period (*j* = 1 ..., 4); β*_k_* is the effect of *k*-th bovine (*k* = 1 …, 4); (δ × R)*_ir_* is the interaction between the treatment and breed; and ҽ*_ijkr_* is the experimental error.

## 3. Results

### 3.1. Nutritional Quality of the Components and the Treatments Evaluated

Table 2 shows the components of the treatments and their nutritional quality. The highest protein contents in the treatments were contributed by cassava leaves (171 g CP/kg DM), followed by concentrate and kikuyu grass, while the chopped cassava root only contributed 33.6 g CP/kg DM to the treatment; thus, the treatments in which 30% of the concentrate was replaced by cassava root (T3) had 15% less protein than the other treatments.

Regarding fiber content, the greatest difference was found in neutral detergent fiber (NDF) values between T2 and T3 (564 vs. 460 kg/d). This same trend was repeated for acid detergent fiber (ADF) content, although the difference was only 53 g/kg DM. The gross energy of the diets ranged from 17.5 to 17.9 MJ/kg DM, while the ash content ranged from 88.1 to 100.1 g/kg. The starch content of cassava root is slightly higher than that of the commercial concentrate (49 and 45.9 g/100 g DM, respectively).

### 3.2. Nutrient and Energy Intake

Table 3 shows nutrient and energy intake and their respective degradability values. Dry matter intake of cows averaged 8.26 kg/day (*p* ≥ 0.05). Protein intake differed among treatments. The highest protein intake was from cassava leaf, followed by concentrate, while chopped cassava root was characterized by a low crude protein content (*p* ≤ 0.05). Daily fiber intake in neutral detergent ranged from 3.64 to 4.80 (*p* ≥ 0.05). Average daily fiber intake in acid detergent was 1.7 kg/d (*p* ≥ 0.05). The average gross energy intake was 149 MJ/d per animal, with a digestibility of approximately 49% (*p* ≥ 0.05).

On average, the mass of degraded dry matter consumed was 4.5 kg (*p* ≥ 0.05), which corresponds to a DM digestibility of 55%. The consumption of compounds such as digested NDF, ADF, and CP was higher for treatments containing all concentrate or the highest proportion of cassava leaves, contrary to what was obtained in treatments where 30% of the concentrate was replaced with cassava root (*p* ≤ 0.05). The digested NDF intake was slightly higher in F1 cows than in Jersey cows (2.07 vs. 1.70 kg/d, respectively) (*p* ≤ 0.05). 

### 3.3. Milk Production and Quality

Table 4 shows the milk production and milk quality of Jersey and F1 cows fed on concentrate, cassava root, and cassava leaves. Milk production did not differ between treatments. However, F1 cows produced more milk than Jersey cows (17.6 vs. 14.7 L/d, respectively) (*p* ≤ 0.05). Fat and protein contents were similar between breeds and between treatments, and their average values were 0.7 and 0.5 kg/d, respectively (*p* ≥ 0.05). The milk urea nitrogen (MUN) content was 15.5 ±2 mg/dL, with no significant differences between treatments. Energy-corrected milk (ECM) production was 1.15 times higher in F1 animals than in Jersey cows (47 vs. 55 kg, respectively) (*p* ≤ 0.05). However, this parameter was not affected by the change in the type of supplementation.

### 3.4. Enteric Methane Emissions 

Table 5 presents enteric CH_4_ emissions in Jersey and F1 cows fed concentrate, cassava root, and cassava leaves. Average enteric CH_4_ emissions per animal ranged from 194 to 234 g/d, with Jersey cows being the lowest producers of this gas (*p* ≤ 0.05). Replacing 30% of the commercial concentrate with cassava root and/or leaves had no effect on CH_4_ emissions (*p* ≥ 0.05). On average, CH_4_ emissions corrected for DM consumption or degraded DM were 26.4 ± 5 and 48.7± 10 g CH_4_/kg, respectively (*p* ≥ 0.05). In the treatments with the substitution of concentrate by root or the mixture of root and cassava leaf, CH_4_ emissions corrected for the consumption of CP digested or utilized by the animal were 25% lower than those in the control diets or treatments with cassava leaves (*p* ≤ 0.05). This same parameter differed between breeds (*p* ≤ 0.05). Gross energy lost as CH_4_ (or the CH_4_ conversion rate, Ym) and CH_4_ emissions associated with ECM production were similar between breeds and between treatments (8.1% and 4.4 g/kg milk, on average, respectively. *p* ≥ 0.05).

### 3.5. Carbon Footprint

Table 6 shows the carbon footprint for a specialized dairy herd under the four different treatments. When relating the carbon footprint to milk production corrected for fat and protein, the treatment where the concentrate was replaced with 30% cassava leaves for the Jersey * Holstein breed led to the least CH_4_ being emitted (1.30 kg CO_2eq_/kg FPCM). For the Jersey breed animals, the treatments with leaves and mixtures of both inputs (leaves and roots) resulted in CH_4_ emissions 6.6% lower than the control treatment (1.54 vs. 1.65 kg CO_2eq_/kg FPCM, respectively). Both trends continue when the carbon footprint is related to weight gains. 

Figure 2 shows the contributions of the different sources to total GHG emissions. In general, the greatest contribution is made by total CH_4_ gas emissions within the farm, which range between 32 and 38.9%, followed by emissions from the manufacture of agrochemicals used for fertilizing pastures (26.4–28.2%) and N_2_O emissions (27.9–31.2%). Of the total GHG emissions calculated, factors such as food manufacturing, fossil fuel burning, and transportation report average values of 6.9, 0.6, and 0.62% respectively. Figure 2 shows that the carbon footprints for cassava-based treatments (T2, T3, and T4) were about 2% lower for the food manufacturing category, compared to the control treatment (T1).

## 4. Discussion

Cassava leaves generated the highest protein intake among the treatments consumed by the animals in this study. Cassava foliage has been reported to be an effective source of bypass protein for feedlot steers [42] and goats [43]. However, the potential of cassava foliage as a source of protein in ruminant feeds has not been fully exploited, probably due to the risk of toxicity. The contents of the hydrogen cyanide (HCN) precursors linamarin and lotaustralin can range from 364 to 964 g/kg depending on variety, age, and plant part [19,44]. Sun drying or ensiling of cassava foliage can reduce HCN toxicity to harmless levels (50 mg/kg) [45], while also improving palatability and prolonging storage time [16]. However, Thang et al. [46] state that high energy intake is needed in cattle diets to counteract the negative effect of HCN. Suharti et al. [19] recommend supplementing cassava leaves with cyanide-degrading bacteria. These latter two alternatives have shown positive results.

In contrast to cassava leaves, the nutritional content of cassava root is characterized by low amounts of protein (2–3%), macronutrients such as fiber (2%), some vitamins, and minerals. However, it has a high calcium, vitamin C, and starch contents [47,48]. The content of this last nutrient in the present investigation was 18% above that reported in the literature (67–81%); this range is a function of the variety of cassava evaluated [49]. Cassava roots also provide the diet with thiamine, riboflavin, carotenoids, minerals, and nicotinic acid, all essential for proper animal metabolism [47,50,51]. Several reports indicate that cassava supplements can replace corn-, cereal-, or tuber-based supplements, maintaining nutritional intake without negative effects on digestibility [19,51,52]. These reports agree with the results of this study, where no differences were observed in DM or organic matter intake, but differences were observed in CP, NDF, and ADF intake when cows received cassava root in their diet (Table 3). 

Regarding DM digestibility, this study found no differences between treatments, although the structural carbohydrate content was different between treatments (Table 3). Digestibility may not have been affected because important system components such as protein and energy supplied the requirements for the correct functioning of ruminal microorganisms. Authors such as Thang et al. [46] and Suharti et al. [19] concluded that the use of cassava with higher levels of protein and metabolizable energy improved microbial protein synthesis and fiber fermentation in diets of cattle that were fed low-quality pasture. However, the inclusion of 25% cassava leaf meal could reduce nutrient digestibility [53]. Additionally, Lunsin et al. [54] reported that supplementation with cassava root hay can improve fiber digestibility, as this type of supplementation significantly improves total bacterial populations of *Ruminococcus flavefaciens* and *Fibrobacter succinogens*. Similarly, Suharti et al. [19] found that supplementation with bitter cassava leaves increased the total bacterial population without an effect on the population of ruminal protozoa. 

The F1 breed exhibited higher total daily milk production compared to the Jersey breed, reflecting inherent genetic disparities between the two. This observation aligns with findings from a study by Coffey et al. [55], where a 305-day assessment revealed that Holstein cows outperformed both the Friesian (4591 kg) and Jersey (4230 kg) breeds, recording a superior production of 5217 kg. Interestingly, our study did not reveal a correlation between dry matter intake and milk yield, contrasting with the results of a meta-analysis by Hristov et al. [56], which showed a moderate linear relationship (R2 = 0.47). This discrepancy might be attributed to the confinement conditions imposed on the animals during our experiment.

The addition of cassava root and leaf did not affect the milk production or milk quality of Jersey and F1 cows (Table 4), confirming other results reported in the literature [57]. Although there were differences in the consumption of NDF, a precursor of milk fat, the production of acetic acid in the rumen may not have differed between diets. Pertiwi et al. [58] concluded that supplementation with 34.5% cassava root husk in Holstein cows positively affected CP and total solids. Their results for fat and milk volume were similar to those of this study. The protein content in milk depends on the energy present in the diet. Therefore, protein contents in milk lower than 3%, values that were not obtained in the present study, are typical of low-energy diets [59]. Likewise, the MUN values obtained range between 14 and 16 mg/dL, indicating an optimal use of nitrogen [60].

In the present study, cassava leaf and root treatments did not significantly change CH_4_ emissions (Table 5), despite differences between digestible intakes of ADF and NDF. The NDF is directly and inversely related to dry matter digestibility, CH_4_ emissions, intake level, and feeding frequency [3,61]. Likewise, the higher contribution of starch present in the root (mainly amylose: 16–18%, and amylopectin: 82–84%) [62] was expected to have an effect on the reduction of CH_4_ emissions [63]. This effect is expected because the digestion of starch in the rumen is decreased, but the digestion of starch in the small intestine is increased, along with a reduced proportion of acetate and butyrate with respect to propionate production [64]. A negative effect of the content of secondary compounds on CH_4_ emissions would also be expected. Why cassava diets improved some digestibility characteristics but did not change CH_4_ emissions remains an open question. Future research could test different cassava sources and higher feed quantities, compared to those used in this study.

In the literature, cassava leaves are reported to contain contents of between 2.06 and 4.36% in tannins as DM in condensed leaves [20,21] and between 1.58 and 1.65 mg/100 g of saponins in bitter cassava leaf meal [65]. These substances lead to decreased CH_4_ production and increased efficiency of microbial protein synthesis [66,67]. However, this study’s results agree with cattle trials where an increase in the level of starch or consumption of secondary compounds in the diet did not affect CH_4_ production per unit DM intake of concentrate or per unit milk produced [19,68]. The molecular weight, chemical structure, and/or amount per kg DM of some of these compounds may explain the lack of effects on CH_4_ emissions [69,70]. According to Binsulong et al. [52], using cassava instead of rice straw in the ration of Holstein Friesian crossbred bulls decreased net CH_4_ emissions due to a reduction in fiber content in the diet.

Net CH_4_ emissions were lower in Jersey than in F1 breeds (Table 5). Authors such as Van Wyngaard et al. [71] reported values of 21.8 kg CH_4_/kg DMI and 6.85% CH_4_ conversion rate (Ym) for multiparous Jersey cows that were 100 (±45.8 SD) days postpartum and had a body weight of 408 (±32.5 SD) kg, whose diet was mainly based on kikuyu grass. These values are slightly lower than those for Jersey cows in our study. In other research using the same technique and kikuyu grass as the base diet, Donneys [72] found that Holstein and Holstein * Simental breed cows of 533 (±81 SD) kg live weight consumed 9.65 kg DM/d and produced 21.65 g CH_4_/kg DMI, with a CH_4_ conversion rate (Ym) of 6.89%.

The carbon footprint corrected for fat and protein content in milk was 1.5 kg CO_2eq_ per kg of FPCM for the four treatments evaluated (Figure 2 and Table 6). These values are within the range reported by Mazzetto et al. [7] for several countries in different continents for non-cassava diets (0.74 and 5.99 kg CO_2eq_ per kg FPCM). In Colombia, some studies report values for non-cassava diets between 2.1 and 4.2 kg CO_2eq_ per kg FPCM and between 9.0 and 18.3 CO_2eq_ per kg of meat for intensive or extensive dual-purpose production systems, respectively [73]. According to Mazzetto et al. [7], the distribution of GHGs in the carbon footprint depends on the type of production system. For example, CH_4_ emissions predominate in pasture-based livestock farming, while N_2_O emissions are highest in intensive or confined livestock farming. In the present study, CH_4_ emissions from enteric fermentation are slightly predominant (35.9%), followed by N_2_O (27%) from manure management, concentrate production, and fertilizer use.

According to Uddin et al. [74], cows have a higher carbon footprint when their diets are low in NDF (19% DM) than when the NDF content is high (24% DM; 1.49 vs. 1.35 kg CO_2eq_/kg FPCM). However, in the present study the cassava root diet provided 8.5% less NDF (460 g/kg DM) than other supplements evaluated and showed a tendency to have lowest carbon footprint with respect to the rest of the treatments evaluated (<3.8%). This can be explained by the differences between the feed resources used [74]. Likewise, the literature reports differences between the carbon footprints of milk for different breeds of cows, where Jersey cows produce 4.4% less carbon than Holstein cows (1.41 vs. 1.47 kg CO_2eq_/kg FPCM, respectively). Uddin et al. [74] attribute this difference to the milk produced by Jersey cows, which has higher fat and protein contents than that of Holstein cows. However, in the present study, this difference was not observed. On the contrary, Jersey cows had a higher carbon footprint than Jersey * Holstein crossbred cows.

In this study we were able to use measured emission factors, as opposed to default emission factors, to calculate the carbon footprint. This aspect is important for several reasons including improved accuracy related to the conditions and management practices of the studied system. These measured emission factors considered important factors, such as the local environment, feed sources, and animal breeds, that can significantly affect CH_4_ emissions. Using measured values leads to more accurate and representative assessments of the carbon footprint.

Default emission factors are often more generalized and may not accurately reflect the specific conditions of this study. The accurate, measured emission factors presented here can be used for developing better accounting (i.e., national GHG inventories), effective policies, and mitigation strategies in Colombia (i.e., nationally determined contributions, NDC). In summary, by using measured emission factors for calculating the carbon footprints of livestock systems in this study, we provide a more accurate, region-specific, and informative basis for understanding and addressing CH_4_ emissions.

## 5. Conclusions

In the treatment evaluated, the greatest amount of protein was provided by cassava leaves, which were also characterized by having five times more fiber content in neutral detergent than roots. These contents influenced nutrient intakes, which were higher for the treatments that included cassava leaves. Lower CH_4_ emissions (g/d) were found in Jersey cows compared to F1 cows (Jersey * Holstein) with no significant statistical differences between treatments in both groups of animals. However, when analyzing CH_4_ emissions corrected for the consumption of CP digested or utilized by the animal, a 25% decrease in emissions was found in treatments with the replacement of concentrate with root or the mixture of root and cassava leaf, compared to control treatments or treatments with cassava leaves (Table 4, CH_4_ (g/d)/CPId (kg)). 

Our results suggest that replacing 30% of the concentrate with cassava leaves and/or roots can slightly reduce the carbon footprint, depending on breed, without affecting the DM consumption, CH_4_ emissions, milk quality, or milk production of cows (Table 5). However, the mixed results found here call for further research on field-based measurements of emissions combined with carbon footprint data. Supplementation with cassava leaves and/or roots can be a nutritionally and environmentally sustainable strategy. Future studies could investigate the effect of higher percentages of cassava in the concentrate fed to dairy cows and the effects of cassava supplementation on Creole breeds with greater hardiness and adaptation to fibrous forages.

## Figures and Tables

**Figure 1 animals-14-00019-f001:**
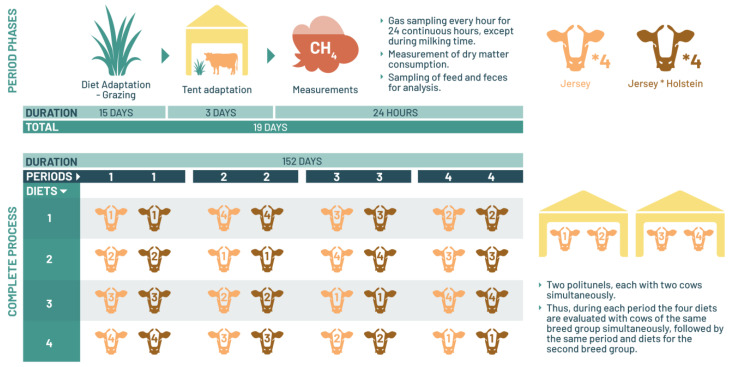
Illustration of the experimental process outlining the study phases for the determination of enteric methane.

**Figure 2 animals-14-00019-f002:**
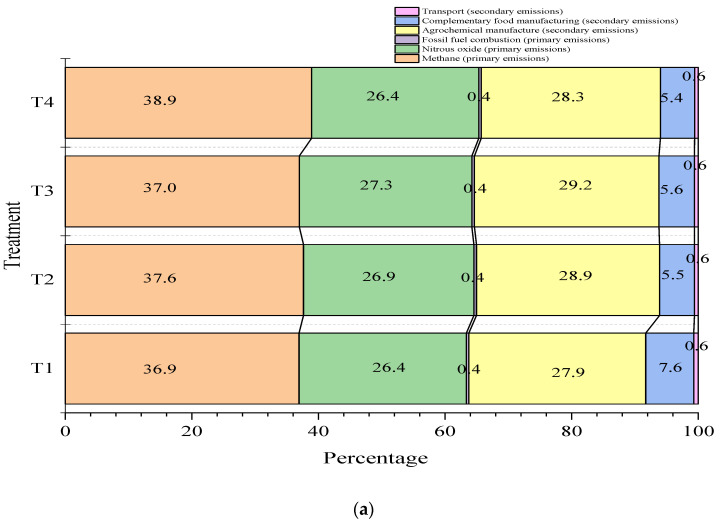
Percentage contribution of greenhouse gas (GHG) sources to the carbon footprint (kg CO_2_eq/kg fat- and protein-corrected milk, FPCM) for various dietary scenarios in Jersey * Holstein F1 (**a**) and Jersey (**b**) breed animals. The carbon footprint values for the control treatment (T1), 30% cassava leaves (T2), 30% root chips, and the mixture of 15% cassava leaves plus 15% chips (T4) were 1.56, 1.54, 1.52, and 1.58 kg CO_2_eq/kg FPCM, respectively.

**Table 1 animals-14-00019-t001:** Compilation of inputs factored into the carbon footprint calculations for the two evaluated dairy herds in this study.

Input	Breed: Jersey * Holstein	Breed: Jersey
Fertilizer 1 ^a^, kg ha^−1^ yr^−1^	1500	1500
Fertilizer 2 ^b^, kg ha^−1^ yr^−1^	1500	1500
Fertilized area, ha	130	70
Petrol, L ha^−1^ yr^−1^	17.7	9.5
Diesel, L ha^−1^ yr^−1^	17.7	9.5

^a^ Fertilizer 1: 25(N): 5(P): 5(K). ^b^ Fertilizer 2: 31(N): 8(P): 8(K).

**Table 2 animals-14-00019-t002:** General nutritional quality of the ingredients and treatments evaluated.

	Ingredients				Treatment			
Items	Cassava Root	Cassava Leaves	Concentrate	Kikuyu	T1	T2	T3	T4
Dry matter (g/kg DM)	860	837	872	141	552	569	570	565
Crude protein (g/kg DM)	33.6	212	171	154	164	171	140	156
Neutral detergent fiber (g/kg DM)	108	587	396	675	518	564	460	503
Acid detergent fiber (g/kg DM)	92.2	471	65.3	370	199	248	195	228
Gross energy (MJ/kg DM)	15.9	19.5	18.3	17.5	17.9	17.8	17.5	17.9
Ashes (g/kg DM)	49.6	76.5	77.3	115	93.9	100.1	88.1	90.6
Starch (g/100 g DM)	49.0	--	45.9	--	--	--	--	--

Legend***:*** T1 = kikuyu grass + concentrate; T2 = kikuyu grass + concentrate + cassava leaves; T3 = kikuyu grass + concentrate + cassava root; T4 = kikuyu grass + concentrate + cassava root + cassava leaves.

**Table 3 animals-14-00019-t003:** Detailed nutritional quality of the components and treatments evaluated, discriminated by cattle breed.

Item	Breed: Jersey				Breed: Jersey * Holstein				MSE	Significance Level (*p*)		
	T1	T2	T3	T4	T1	T2	T3	T4		Treatments	Breed	T*B
**Intake**												
Dry matter (kg/d)	8.21	8.11	7.85	7.93	8.89	8.03	8.56	8.51	0.946	0.488	0.176	0.913
Dry matter (kg/MW)	9.03	8.90	8.63	8.71	8.41	7.60	8.10	8.04	0.990	0.496	0.272	0.932
Organic matter (kg/d)	7.47	7.33	7.19	7.24	8.08	7.24	7.83	7.76	0.826	0.540	0.172	0.929
Crude protein (g/d)	1.40 ^a^	1.44 ^a^	1.15 ^b^	1.29 ^a,b^	1.51 ^a^	1.44 ^a^	1.27 ^b^	1.39 ^a,b^	0.156	0.012	0.159	0.914
NDF (kg/d)	4.32	4.51	3.64	4.04	4.80	4.46	4.14	4.45	0.669	0.074	0.169	0.999
ADF (kg/d)	1.47	1.99	1.37	1.66	1.71	1.95	1.60	1.86	0.329	0.559	0.164	0.883
**Digestible nutrient intake**												
Dry matter (kg/d)	4.49	4.42	4.25	4.31	4.92	4.37	4.71	4.68	0.613	0.787	0.189	0.8627
Organic Matter (kg/d)	4.19	4.07	4.03	4.02	4.63	4.08	4.49	4.48	0.514	0.635	0.073	0.752
Crude protein (g/d)	748 ^a^	750 ^a^	526 ^b^	642 ^a,b^	702 ^a^	691 ^a^	483 ^b^	557 ^a,b^	135.3	0.009	0.236	0.989
NDF (kg/d)	1.86 ^a,B^	2.08 ^a,B^	1.25 ^b,B^	1.62 ^a,b,B^	2.33 ^a,A^	2.18 ^a,A^	1.72 ^b,A^	2.04 ^a,b,A^	0.397	0.011	0.013	0.756
ADF (kg/d)	0.16 ^c^	0.54 ^a^	0.11 ^c^	0.37 ^b^	0.27 ^c^	0.44 ^a^	0.27 ^c^	0.38 ^b^	0.124	0.001	0.541	0.442
**Energy consumption (MJ/d*)***												
Gross energy	149	149	139	149	161	147	152	1531	16.97	0.702	0.177	0.821
Digestible energy	76.5	77.6	70.3	70.3	82.5	75.4	76.8	75.1	10.66	0.570	0.323	0.806

Legend: ^a,b,c^ Lowercase letters indicate differences between treatments within each group (*p* ≤ 0.05). ^A,B^ Capital letters indicate differences between groups (*p* ≤ 0.05). MSE: Mean square error; T1 = kikuyu grass + concentrate, T2 = kikuyu grass + concentrate + cassava leaves, T3 = kikuyu grass + concentrate + cassava root, T4 = kikuyu grass + concentrate + cassava leaves + cassava root; MW = Metabolic Weight, NDF = neutral detergent fiber, ADF = acid detergent fiber, T*B = treatment-breed interaction.

**Table 4 animals-14-00019-t004:** Milk production and milk quality for Jersey and F1 cows under varying nutritional treatments comprising concentrate, cassava root, and cassava leaves.

Item	Breed: Jersey				Breed: Jersey * Holstein				MSE	Significance Level (*p)*		
	T1	T2	T3	T4	T1	T2	T3	T4		Treatments	Breed	T*B
Quantity (L/d)	15.0 ^B^	12.3 ^B^	15.5 ^B^	16.0 ^B^	18.8 ^A^	17.3 ^A^	17.5 ^A^	17.0 ^A^	3.093	0.528	0.014	0.583
Fat (kg/d)	0.62	0.57	0.67	0.69	0.75	0.83	0.70	0.68	0.167	0.998	0.087	0.637
Protein (kg/d)	0.47	0.39	0.50	0.50	0.56	0.52	0.55	0.51	0.103	0.358	0.053	0.405
Solids (kg/d)	1.09	0.95	1.18	1.19	1.31	1.35	1.25	1.19	0.259	0.977	0.062	0.512
MUN (mg/dL)	15.5	16.6	14.6	16.1	14.9	15.5	16.2	14.6	1.605	0.739	0.456	0.235
ECM (kg/d)	47.8 ^B^	40.2 ^B^	51.2 ^B^	51.5 ^B^	57.6 ^A^	54.9 ^A^	55.6 ^A^	52.1^A^	9.894	0.634	0.046	0.501

Legend ^A,B^ Capital letters indicate differences between breeds (*p* ≤ 0.05). SEM: Mean square error; T1 = kikuyu grass + concentrate, T2 = kikuyu grass + concentrate + cassava leaves, T3 = kikuyu grass + concentrate + cassava root; T4 = kikuyu grass + concentrate + cassava leaves + cassava root. T*B = Treatments-breed interaction. MUN = Milk ureic nitrogen. ECM = energy-corrected milk.

**Table 5 animals-14-00019-t005:** Methane emissions from Jersey and F1 cows fed on concentrate, cassava root and cassava leaves, and cassava root or leaves.

Item	Breed: Jersey				Breed: Jersey * Holstein				MSE	Significance Level (*p)*		
	T1	T2	T3	T4	T1	T2	T3	T4		Treatments	Breed	T*B
Methane (g/d)	187 ^B^	193 ^B^	193^B^	205 ^B^	234^A^	232 ^A^	224 ^A^	250 ^A^	32.46	0.657	0.002	0.959
Methane (g/d)/DMI (kg)	23.5	23.8	25.2	25.7	26.4	30.5	26.4	30.0	5.276	0.693	0.055	0.763
Methane (g/d)/DMId (kg)	43.4	43.7	46.8	47.6	47.8	57.4	48.1	55.1	10.56	0.676	0.084	0.685
Methane (g/d)/OMId (kg)	46.5	47.3	49.1	50.9	50.7	60.4	50.4	57.6	10.72	0.643	0.109	0.732
Methane (g/d)/NDFId (kg)	112	93	189	138	102	128	136	151	49.58	0.103	0.842	0.362
Methane (g/d)/CPId (kg)	0.27 ^b,B^	0.26 ^b,B^	0.39 ^a,B^	0.32 ^a,B^	0.35 ^b,A^	0.38 ^b,A^	0.49 ^a,A^	0.50 ^a,A^	0.105	0.049	0.003	0.661
Ym (%)	7.20	7.22	7.91	7.93	8.11	9.23	8.29	9.28	1.61	0.696	0.057	0.773
Methane (g)/Milk (kg)	12.7	16.7	13.1	13.5	12.8	13.7	13.4	14.7	3.857	0.614	0.803	0.727
Methane (g)/ECM kg	3.97	5.13	3.96	4.30	4.19	4.35	4.23	4.83	1.21	0.625	0.874	0.713

Legend: ^a,b^ Lowercase letters indicate differences between treatments within each breed (*p* ≤ 0.05). ^A,B^ Capital letters indicate differences between breeds (*p* ≤ 0.05). MSE: Mean square error; T1 = kikuyu grass + concentrate; T2 = kikuyu grass + concentrate + cassava leaves; T3 = kikuyu grass + concentrate + cassava root; T4 = kikuyu grass + concentrate + cassava leaves + cassava root; T*B =Treatment-breed interaction; DMI = Dry Matter Intake; DMId = intake of degraded dry matter; OMId = intake of degraded organic matter; NDFId = intake of degraded neutral detergent fiber, CPId = intake of degraded crude protein. Ym = fraction of gross energy consumed by an animal that is converted to methane, ECM = energy-corrected milk.

**Table 6 animals-14-00019-t006:** Milk and meat carbon footprint comparison for specialized dairy cows, fed kikuyu grass and supplemented with either commercial concentrate, cassava root, or cassava leaves.

	Jersey * Holstein				Jersey			
Item	T1	T2	T3	T4	T1	T2	T3	T4
Carbon footprint (kgCO_2_eq)/FPCM (kg)	1.35	1.30	1.34	1.42	1.65	1.75	1.54	1.54
Carbon footprint (kgCO_2_eq)/LWG (kg)	10.6	10.2	10.7	11.5	12.3	13.4	11.4	11.3

Legend: T1 = kikuyu grass + concentrate; T2 = kikuyu grass + concentrate + cassava leaves; T3 = kikuyu grass + concentrate + cassava root; T4 = kikuyu grass + concentrate + cassava leaves + cassava root; FPCM: milk corrected for fat (3.5%) and protein (3.3%) content; LWG: liveweight gain.

## Data Availability

All authors ensure that all data and materials support the findings and comply with field standards.

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
