# Peer review of "Methane Emission, Carbon Footprint and Productivity of Specialized Dairy Cows Supplemented with Bitter Cassava (Manihot esculenta Crantz)"

_animals, 2023, doi:10.3390/ani14010019_

Round 1

Reviewer 1 Report

Comments and Suggestions for Authors

Dear authors. Thanks for considering this journal for the publishing of your results. I want to highlight the importance to develop field trials that confront many and varied information with the real conditions of livestock systems, especially those in tropic environments.  

I hope the next comments and questions improve the clarity of your message as well as this manuscript’ scientific quality.

General comments

When looking into the objective of this work (determine the effect of supplementation with cassava leaves and roots on CH4 emissions, carbon footprint, and some productive parameters), the majority of results were not significant from an inferential point of view. This is, the four diets showed a general trend to not change the parameters estimated. But another important component arose when authors include the breed component as an independent variable. From my point of view, it was difficult to understand and seems more like a strategy to quest statistical differences but not to discuss the biological reasons behind the main results that are clearly the lack of differences between diets (T1, T2, T3, T4) in most of the studied parameters. Breed, as an influential factor, was not part of the introductory section of the manuscript and only appears later in the m&m section. Further confusion was added since authors argue in line 121 that experimental groups were categorized according with the breed, but the general idea in the manuscript indicates that the original categorization was the four diets administered to the cows and this was reinforced in the conclusion (lines 442-443). In this case, breed would help as an explanatory factor or as a covariable. In this sense, my invitation to authors is to focus on those outcomes and discussing the biological processes leading to not detect changes in the productive, physiological, and environmental measured variables.

The 30% threshold established for the present work seems somewhat arbitrary. There is previous data/trials leading authors to establish it? If so, it would be important to add this information in the introduction and/or the m&m sections

The section 2.4 is difficult to understand. After reading it many times, I consider that, as currently redacted, it could be challenging to be clear and reproducible. The suggestion thus is –in addition to a grammar modification– to elaborate a figure illustrating the experimental process. This will help your future readers.

A similar comment goes for 2.5. In this case, a photo (or grafic) of polytunnel functioning could add a better idea and understanding of your measurements.

In the discussion (Lines 337-340 and 351-353) authors bring information on parameters that were not measured in the present study. This information, albeit important, has no relation with this work and in my opinion should be removed.

The same comment from above goes for Lines 384-386. The present work did not assess the presence of PSM and discussions on this issue could be interpreted as apart from this work’ measurements

The arguments in Lines 413-415 are challenging from a statistical point of view. Although there was a difference (probably too short [3.8 percentual points]), it was not statistically significant. However, as currently redacted, one could think an important change. Please be more conservative in discussing those outcomes.

The same comment for the conclusion (lines 449-451)

Specific comments

Title: Please verify whether the complete name is in italic. I think the ‘crantz’ should be written regularly.

Title: Maybe ‘emissions’ goes better in singular (to maintain the pattern with ‘footprint’ and ‘productivity’)

The introductory phrase in abstract and simple summary is important but it is not linked with an upcoming idea. Please re-write or remove

The scientific name of cassava has to be established in the abstract

L37: Although there were no differences in CH4 emission, a P<0.05 appear in the abstract. Please correct

L54: Please add a reference for that phrase

L56: Since they appear once in the text, the acronyms could be removed

L69: Please establish the scientific name with the corresponding botanic family and first describer: Manihot esculenta Crantz. (Euphorbiaceae)

L71-72: The phrase is difficult to understand

L80: I congratulate the authors for perform the first study in this area. However, the redaction might be changed or maybe declare this strength beginning the discussion

L90: It is important to add a reference on the weather/climatological conditions

L93-94: The basic nutritional information of the commercial concentrate (CP, ME, FDN, FDA) should be provided in the m&m section

L100: ….and supply to the animals in the tested farm

Please consider reverting the order. I think that “experimental design and animal management” goes first (2.3) and as a consequence “nutritional quality of diets” goes later

L118: Neenich’s equation should be established and explained

L181: Please explain the meaning of colanta.

L181: Colanta’s®

L207: What is the meaning of ‘Tier’

L320: It would be more recommendable to manage mg/kg instead ppm

L323: “recommend additional energy intake is needed” is hard to understand

L329 “was 18% above”

L337-340: But these outcomes are not applicable under the present research. It is important information with no comparative aim.

L351-353: The same previous comment. Ruminal populations were not studied, and any comparison seems questionable. In addition, the phrase is difficult to understand in its actual redaction.

L363: Their results…

L388-390: The redaction makes difficult to understand those outcomes from Binsulong

Comments on the Quality of English Language

In my opinion, the manuscript needs English editing. There is a characteristic ‘footprint’ from the Spanish language throughout the text. Although it is not bad written, the fluency and message clarity might be compromised under an academic point of view and an English-reader audience.  I think it will help this manuscript.

Author Response

Comments from the editors and reviewers and our responses:

Methane emission, carbon footprint and productivity of specialty dairy cows supplemented with bitter cassava (Manihot esculenta Crantz)

by Molina-Botero et al.,

We express our gratitude to the editor and reviewers for their valuable suggestions. In this document, we outline the modifications implemented in response to the feedback provided.

Revisor 1

Comment

Response

General comments

When looking into the objective of this work (determine the effect of supplementation with cassava leaves and roots on CH4 emissions, carbon footprint, and some productive parameters), the majority of results were not significant from an inferential point of view. This is, the four diets showed a general trend to not change the parameters estimated. But another important component arose when authors include the breed component as an independent variable. From my point of view, it was difficult to understand and seems more like a strategy to quest statistical differences but not to discuss the biological reasons behind the main results that are clearly the lack of differences between diets (T1, T2, T3, T4) in most of the studied parameters. Breed, as an influential factor, was not part of the introductory section of the manuscript and only appears later in the m&m section. Further confusion was added since authors argue in line 121 that experimental groups were categorized according with the breed, but the general idea in the manuscript indicates that the original categorization was the four diets administered to the cows and this was reinforced in the conclusion (lines 442-443). In this case, breed would help as an explanatory factor or as a covariable. In this sense, my invitation to authors is to focus on those outcomes and discussing the biological processes leading to not detect changes in the productive, physiological, and environmental measured variables.

The 30% threshold established for the present work seems somewhat arbitrary. There is previous data/trials leading authors to establish it? If so, it would be important to add this information in the introduction and/or the m&m sections

The section 2.4 is difficult to understand. After reading it many times, I consider that, as currently redacted, it could be challenging to be clear and reproducible. The suggestion thus is –in addition to a grammar modification– to elaborate a figure illustrating the experimental process. This will help your future readers.

A similar comment goes for 2.5. In this case, a photo (or grafic) of polytunnel functioning could add a better idea and understanding of your measurements.

In the discussion (Lines 337-340 and 351-353) authors bring information on parameters that were not measured in the present study. This information, albeit important, has no relation with this work and in my opinion should be removed.

The same comment from above goes for Lines 384-386. The present work did not assess the presence of PSM and discussions on this issue could be interpreted as apart from this work’ measurements

The arguments in Lines 413-415 are challenging from a statistical point of view. Although there was a difference (probably too short [3.8 percentual points]), it was not statistically significant. However, as currently redacted, one could think an important change. Please be more conservative in discussing those outcomes.

The same comment for the conclusion (lines 449-451)

Thank you for your valuable comments. We have addressed each of your points as follows:

In the introduction, we have incorporated a new paragraph (L9-L103) that delves into the breed factor.

Within the Materials and Methods section, we have provided further clarification on why 30% was determined as the optimal percentage for replacing concentrate with cassava supplements (L124-L130). Additionally, we included a figure (L126) to enhance the understanding of the experimental process. The added text now explains the rationale behind determining the maximum replacement percentage based on a prior test involving 10 randomly selected animals.

In the discussion section, we have removed references to parameters that were not measured (L438-441 and L456-L459).

Minor adjustments have been made to the wording in order to avoid conveying a significant difference message when statistical significance was not observed.

Title: Please verify whether the complete name is in italic. I think the ‘crantz’ should be written regularly.

The modification has been applied to the title, summary, and the Materials and Methods section (L3, L17, L28, L78).

Title: Maybe ‘emissions’ goes better in singular (to maintain the pattern with ‘footprint’ and ‘productivity’)

The suggestion has been implemented in the title (L1).

The introductory phrase in abstract and simple summary is important but it is not linked with an upcoming idea. Please re-write or remove

Both sentences were changed accordingly (L16-17 and L27-28).

The scientific name of cassava has to be established in the abstract

The addition has been made in both sections (L17 and L28).

L37: Although there were no differences in CH4 emission, a P<0.05 appear in the abstract. Please correct

This was corrected in the abstract section (L37)

L54: Please add a reference for that phrase

For the sentence "However, its permanence in the atmosphere is lower", a reference was added (L46-47).

L56: Since they appear once in the text, the acronyms could be removed

The abbreviation CF (carbon footprint was deleted (L51).  

L69: Please establish the scientific name with the corresponding botanic family and first describer: Manihot esculenta Crantz. (Euphorbiaceae)

This is described in the introduction on line 80.

L71-72: The phrase is difficult to understand

This sentence was corrected (L81-83).

L80: I congratulate the authors for perform the first study in this area. However, the redaction might be changed or maybe declare this strength beginning the discussion

The phrase was deleted as redundant (L92-95).

L90: It is important to add a reference on the weather/climatological conditions

This was added to the text (L115).

L93-94: The basic nutritional information of the commercial concentrate (CP, ME, FDN, FDA) should be provided in the m&m section

This information was provided in the results section in section 3.1 where all the nutritional composition of the feeds provided are described (Table 2).

L100: ….and supply to the animals in the tested farm

This was corrected (L134)

Please consider reverting the order. I think that “experimental design and animal management” goes first (2.3) and as a consequence “nutritional quality of diets” goes later

Thanks for the comment, the order of these sections was changed.

L118: Neenich’s equation should be established and explained

The equation was added in the text (L180-183)

L181: Please explain the meaning of colanta.

Colanta corresponds to the company name

L181: Colanta’s®

The symbol ® was added to the name Colanta (L260).

L207: What is the meaning of ‘Tier’

Tier means level, this term is widely used in the area. We do not think we need to explain it.

L320: It would be more recommendable to manage mg/kg instead ppm

This was changed in the text (L420).

L323: “recommend additional energy intake is needed” is hard to understand

This was changed in the text to the following sentence: "recommend high energy intake is needed in cattle diets to counteract the negative effect of HCN" (L423-424)

L329 “was 18% above”

The word "is" was changed to "was" (L430).

L337-340: But these outcomes are not applicable under the present research. It is important information with no comparative aim.

This sentence was deleted (L438-441)

L351-353: The same previous comment. Ruminal populations were not studied, and any comparison seems questionable. In addition, the phrase is difficult to understand in its actual redaction.

This phrase was deleted (L456-460)

L363: Their results…

This was corrected in the text (L475)

L388-390: The redaction makes difficult to understand those outcomes from Binsulong

This was corrected in the text (501-502)

Other changes made

- Language correction

- Unify terms between treatments and diets throughout the document.

- Document formatting changes

Reviewer 2 Report

Comments and Suggestions for Authors

Ms. Ref. No.: animals-2701563

Title: “Methane emissions, carbon footprint and productivity of specialty dairy cows supplemented with bitter cassava (Manihot esculenta crantz)

Animals

General comments

I have had the opportunity to review the manuscript. The manuscript is interesting and is in line with the topic of the journal, however it needs more details in the methodology some points in the discussions. In particular, more details must be provided on the methodology used and on the primary information for LCA.

It is necessary to explain in the discussion how it is possible that animals have different live weights and equal intakes given the same treatment without any significant difference, with significantly greater milk production.

Below my considerations:

Introduction

 It is necessary to include an introductory part also on the carbon footprint in the dairy livestock sector, we recommend reading and cite the following articles:

https://doi.org/10.1016/j.livsci.2023.105273

https://doi.org/10.3390/su12052128

10.3168/jds.2022-22804

L 51 articles under review cannot be cited, articles in press and accepted can be cited, delete.

Material and methods

 L 190 articles under review cannot be cited, articles in press and accepted can be cited, delete. Cite the sources of the IPCC equations

L 192-214 it is necessary to provide a summary table with all the primary information of the two systems being compared for LCA study, such as diesel consumption, electricity consumption, etc.

Comments on the Quality of English Language

Extensive editing of English language required

Author Response

Comments from the editors and reviewers and our responses:

Methane emission, carbon footprint and productivity of specialty dairy cows supplemented with bitter cassava (Manihot esculenta Crantz)

by Molina-Botero et al.,

We express our gratitude to the editor and reviewers for their valuable suggestions. In this document, we outline the modifications implemented in response to the feedback provided.

Revisor 2

Comment

Response

General comments

I have had the opportunity to review the manuscript. The manuscript is interesting and is in line with the topic of the journal, however it needs more details in the methodology some points in the discussions. In particular, more details must be provided on the methodology used and on the primary information for LCA.

It is necessary to explain in the discussion how it is possible that animals have different live weights and equal intakes given the same treatment without any significant difference, with significantly greater milk production.

 Below my considerations:

Thank you for your comments, below, we provide answers to them:

- We add the primary information used for the calculation of the Carbon footprint (L285-293).

- This was included in the discussion (L461-469).

 Introduction

 It is necessary to include an introductory part also on the carbon footprint in the dairy livestock sector, we recommend reading and cite the following articles:

 https://doi.org/10.1016/j.livsci.2023.105273

https://doi.org/10.3390/su12052128

10.3168/jds.2022-22804

In the introduction a paragraph was included where the topic of carbon footprint in the dairy cattle sector is deepened (L53-58).

L 51 articles under review cannot be cited, articles in press and accepted can be cited, delete.

This was changed by another reference already published (reference #5) (L781-783).

L 190 Cite the sources of the IPCC equations

References describing the equations used for carbon footprint calculations were added (L273).

 L 192-214 it is necessary to provide a summary table with all the primary information of the two systems being compared for LCA study, such as diesel consumption, electricity consumption, etc.

We added the primary data used for carbon footprint calculations (L275-285).

Other changes made

- Language correction

- Unify terms between treatments and diets throughout the document.

- Document formatting changes

Reviewer 3 Report

Comments and Suggestions for Authors

It is a nice manuscript.

Items to be changed:

1) 216 A double change-over experimental design was used: This is also mentioned in line 128 already.

2) Fig. 1: Use a larger font for the legend 

3) Line 458, 6. Patents: remove if not relevant 

4) Line 476, Acknowledgments: We recognize DeepL tool for the translation of this text: you do not cite any similar work? Pls. add that source and make sure that any self-citations are properly done.

Specialty dairy systems

Author Response

Comments from the editors and reviewers and our responses:

Methane emission, carbon footprint and productivity of specialty dairy cows supplemented with bitter cassava (Manihot esculenta Crantz)

by Molina-Botero et al.,

We express our gratitude to the editor and reviewers for their valuable suggestions. In this document, we outline the modifications implemented in response to the feedback provided.

Revisor 3

Comment

Response

1) 216 A double change-over experimental design was used: This is also mentioned in line 128 already

This was removed from the "data analysis" section (L309-310).

2) Fig. 1: Use a larger font for the legend 

This was corrected in the text (L407-412)

3) Line 458, 6. Patents: remove if not relevant 

This word was deleted (L573)

4) Line 476, Acknowledgments: We recognize DeepL tool for the translation of this text: you do not cite any similar work? Pls. add that source and make sure that any self-citations are properly done.

We believe that as a tool used in the translation of the text, we must reference it.

Other changes made

- Language correction

- Unify terms between treatments and diets throughout the document.

- Document formatting changes

Round 2

Reviewer 1 Report

Comments and Suggestions for Authors

-

Comments on the Quality of English Language

-

Reviewer 2 Report

Comments and Suggestions for Authors

ok.